# Development and validation of a quantitative, non-invasive, highly sensitive and specific, electrochemical assay for anti-SARS-CoV-2 IgG antibodies in saliva

**Samantha H. Chiang**[1]**, Michael Tu**[3]**, Jordan Cheng**[1]**, Fang Wei**[1]**, Feng Li**[1]**, David Chia**[2]**, Omai Garner**[2]**, Sukantha Chandrasekaran**[2]**, Richard Bender**[3]**, Charles M. Strom**[3]**\*, David T. W. Wong**[1]

1 School of Dentistry, University of California, Los Angeles, CA, United States of America, 2 Department of Pathology and Laboratory Medicine, University of California David Geffen School of Medicine, Los Angeles, CA, United States of America, 3 Liquid Diagnostics, LLC, San Clemente, CA, United States of America

\* strom@liquid-dx.com

**Data Availability Statement:** Data is available on figshare: https://figshare.com/projects/Development_and_validation_of_a_quantitative_

## Abstract

Amperial™ is a novel assay platform that uses immobilized antigen in a conducting polymer gel followed by detection via electrochemical measurement of oxidation-reduction reaction between $H_2O_2$/Tetrametylbenzidine and peroxidase enzyme in a completed assay complex. A highly specific and sensitive assay was developed to quantify levels of IgG antibodies to SARS-CoV-2 in saliva. After establishing linearity and limit of detection we established a reference range of 5 standard deviations above the mean. There were no false positives in 667 consecutive saliva samples obtained prior to 2019. Saliva was obtained from 34 patients who had recovered from documented COVID-19 or had documented positive serologies. All of the patients with symptoms severe enough to seek medical attention had positive antibody tests and 88% overall had positive results. We obtained blinded paired saliva and plasma samples from 14 individuals. The plasma was analyzed using an EUA-FDA cleared ELISA kit and the saliva was analyzed by our Amperial™ assay. All 5 samples with negative plasma titers were negative in saliva testing. Eight of the 9 positive plasma samples were positive in saliva and 1 had borderline results. A CLIA validation was performed as a laboratory developed test in a high complexity laboratory. A quantitative non-invasive saliva based SARS-CoV-2 antibody test was developed and validated with sufficient specificity to be useful for population-based monitoring and monitoring of individuals following vaccination.

## Introduction

A novel corona virus, severe acute respiratory syndrome coronavirus 2 (SARS-CoV-2), has caused a global pandemic causing major disruptions world-wide [1]. Multiple high-throughput PCR based tests have been developed that are reasonably sensitive and specific, however the same cannot be said for antibody testing, prompting The Center for Disease Control (CDC) to

non-invasive_highly_sensitive_and_specific_
electrochemical_assay_for_anti-SARS-CoV-2_
IgG_antibodies_in_saliva/112851.

**Funding:** DW is supported by U54HL119893, UCLA W.M. Keck Foundation Research Award Program. SC is supported by F30DE027615. This study was partially funded by Liquid Diagnostics, LLC (LD). The funder provided reimbursement to MT as a paid consultant. This author contributed to this study by performing some experiments and in manuscript preparation. He did not contribute to the decision to publish, data collection or analysis.

**Competing interests:** CS in an unpaid CEO of LD. CS, MT, RB, and DW are equity holders in LD. LD is the exclusive license holder for the Amperial™ technology from the University of California and hopes to commercialize products based on this technology. This does not alter our adherence to PLOS ONE policies on sharing data an materials.

issue guidelines entitled "Interim Guidelines for COVID-19 Antibody Testing" [2]. This publication describes the variability of in-home antibody tests and the lack of specificity required to make home-based antibody testing a valuable tool for epidemiologic surveillance.

Having a reliable self-collection antibody test may be of enormous help in epidemiologic studies of background immunity, testing symptomatic individuals without RNA based testing during their acute illness, and screening health care providers and first responders to establish prior COVID-19 infection. Such a test may also be valuable in following vaccinated patients to assess the kinetics of anti-SARS-CoV-2 antibody production following inoculation. Multiple serological tests based on serum or plasma have been developed and marketed, with ELISA and lateral flow methods predominating. However, many methods suffer from low sensitivities and specificities [2–6].

Antibodies begin appearing in the first week following the development of symptoms. IgG, IgM, and IgA are detectable with IgA appearing somewhat earlier than IgG and IgM. Most patients seroconvert by 2 weeks following symptoms. Unlike IgA and IgM, IgG persists for several months following infection [7–9].

In a published study of 1,797 Icelandic individuals recovered from qPCR documented COVID-19 disease, 91% were IgG seropositive and antibody levels remained stable for 4 months after initial symptoms [10]. Notably 2.3% of individuals quarantined due to exposure but untested for virus, with negative qPCR results, tested positive for IgG antibodies. Of 18,609 patients who were both unexposed and asymptomatic, the seropositivity rate was 0.3% [11].

Since health care systems are burdened with care for COVID-19 patients, having a test that does not require phlebotomy would be extremely beneficial. To that end, investigations have been carried out using home finger prick blood sampling and even some home blood spot testing lateral flow strips [5–7]. However, home finger stick is invasive and not acceptable to some individuals, and requires a health care professional to administer the test to vulnerable individuals such as the elderly and children. In addition, home blood collection tests are less accurate than phlebotomy, with specificities less than 98%. In a low prevalence disease, the positive predictive value for a test with 98% specificity is less than 50% [7, 11].

Saliva is an oral fluid that is obtained easily and non-invasively. Proteomic studies show that the immunoglobulin profile in saliva is nearly identical to that of plasma [12]. Therefore, saliva is an excellent medium for COVID-19 antibody measurement. There are several commercially available collection devices to facilitate saliva collection, stabilization of IgG, and transport.

A recently published study demonstrated excellent correlation between levels of COVID-19 antibodies in serum and saliva [13]. In order to be useful in population-based screening and to determine individual immunity in exposed populations, a SARS-CoV-2 antibody test must be highly specific because of the low seroprevalence rate in the population [2, 14]. In addition, the ability to quantify antibody levels is important for vaccine development and in monitoring for waning immunity [2, 14]. The only published saliva based assay for SARS-CoV-2 antibodies had only 89% sensitivity with 98% specificity [13], leading to a positive predictive value of only 49% in a population with a 2% prevalence of COVID-19 exposure.

Our goal was to develop a non-invasive saliva based quantitative test for COVID-19 antibodies with exquisite sensitivity. We reviewed existing literature to find the SARS-CoV-2 antigen domain with the highest specificity and the ability to distinguish between the COVID-19 virus and other related Coronaviruses. The S1 domain is the most specific in terms of cross reactivity with other Corona and other respiratory viruses. As recombinant S1 antigen is readily available from at least 2 vendors, we chose the S1 antigen for our assay development.

Levels of IgM and IgA deteriorate rapidly following recovery from COVID-19 infection; IgG levels remain detectable for several weeks to months [10]. Since the intended use of our

assay is for population-based screening and vaccine efficacy monitoring, we chose to assay IgG only.

The Amperial™ technology, formerly known as Electric Field Induced Release and Measurement (EFIRM™), is a novel platform capable of performing quantitation of target molecules in both blood and saliva [15]. The device works by immobilizing capture moieties on the surface of an electrode structure for capturing target analytes and then quantifying the target analyte through electrochemically measuring oxidation-reduction between a hydrogen peroxide and tetramethylbenzidine substrate and peroxidase enzyme in a completed assay sandwich. The assay takes place on electrodes packaged in the format of a traditional 96-well microtiter plate, making the assay technique highly compatible and scalable with existing lab liquid handling instruments.

We developed quantitative Amperial™ assays for IgG, IgM, and IgA antibodies to the S1 spike protein antigen of SARS-CoV-2. This test is highly sensitive (>88%) and specific (>99.85%) for patients with COVID-19 infections and correlates well with plasma ELISA analysis. The unique assay described in this article is completely non-invasive, allows home-collection, is quantitative, and has shown no false positives in 667 unexposed individuals, leading to a specificity of at least 99.6%. The assay has strong utility for clinical laboratories as it does not require purification/extraction of the saliva specimen, but the sample can simply be pipetted out of the collection device, diluted, and pipetted to the assay plate. The turnaround time of the assay is also fast, requiring less than 1 hour for a complete assay to be run.

The widespread use of this test may be of great value in identifying individuals with prior exposure to SARS-CoV-2, to follow patients longitudinally to determine the kinetics of diminishing antibody concentration, and may be of special value in the longitudinal monitoring of vaccinated individuals to assess continued serologic immunity.

## Materials and methods

The schematic of the Amperial™ SARS-CoV-2 IgG antibody is shown in Fig 1. The principle of the Amperial™ platform is that a biomolecule (in this case SARS-CoV-2 Spike protein S1 antigen) is added to a liquid pyrrole solution that is then pipetted into the bottom of microtiter wells containing a gold electrode at the bottom of each well. After the solution is added to each well, the plate is placed into the Amperial™ Reader and subjected to an electric current leading to polymerization. This procedure results in each well becoming coated with a conducting polymer gel containing the S1 antigen. Following the polymerization, diluted saliva, plasma, or serum is added to the well. Specific anti-S1 antibodies bind to the S1 antigen in the polymer. After rigorous washing procedures, the bound antibody is detected by using biotinylated anti-human IgG and then the signal is amplified by a standard streptavidin / horseradish peroxidase reaction that produces an electric current measured by the Amperial™ Reader in the nanoampere (nA) scale. The instrument is capable of accurately measuring current in the picoampere (pA) range, so the measurement is well within the ability of the instrument [13, 14, 16, 17]. The measurement of current rather than optical absorbance, as is done in the typical ELISA, has two important advantages over standard ELISA. Firstly, it allows precise quantitation of

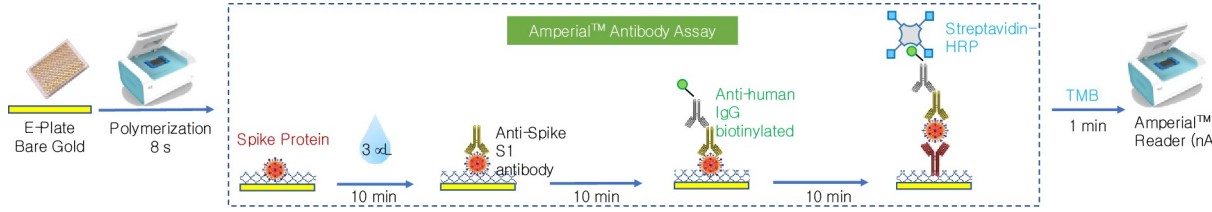

**Fig 1. Schematic of the Amperial™ saliva anti-SARS-CoV-2 IgG assay.** See methods for description.

the amount of bound antibody and secondly, the measurement of current rather than optical absorbance allows increased sensitivity. Since antibody levels in saliva are lower than in plasma [13, 16], this increased sensitivity is crucial. The precise details of the assay are described in the next paragraph.

COVID-19 Spike-1 Antigen (Sanyou-Bio, Shanghai, China) was diluted to a concentration of 6.25 μg / mL, added to each well of the microtiter plate, and co-polymerized with pyrrole (Sigma-Aldrich, St. Louis, MO) onto the bare gold electrodes by applying a cyclic square wave electric field at 350 mV for 1 second and 1100 mV for 1 second. In total, polymerization proceeded for 4 cycles of 2 seconds each. Following this electro-polymerization procedure, 6 wash cycles were performed using 1x PBS with 0.05% Tween-20 (PBS-T) using a 96-channel Biotek 405LS plate washer programmed to aspirate and dispense 400 μL of solution per cycle.

Following the application of the polymer layer, 30 μL of saliva diluted at a 1:10 ratio in Casein/PBS (Thermo-Fisher, Waltham, MA) was pipetted into each well and incubated for 10 minutes at room temperature. Unbound components were removed by performing 6 wash cycles of PBS-T using the plate washer.

Biotinylated anti-human IgG secondary antibody (Thermofisher, Waltham, MA) at a stock concentration of 1.5 mg / mL was diluted 1:500 in Casein/PBS and 30 μL pipetted to the surface of each well and incubated for 10 minutes at room temperature followed by 6 wash cycles using PBS-T. Subsequently, 30 μL of Poly-HRP80 (Fitzgerald Industries, Acton, MA) at a stock concentration of 2 μg / mL was diluted 1:25 in Casein/PBS, added to the wells, and incubated at 10 minutes at room temperature. Following a final wash using 6 cycles of PBS-T, current generation is accomplished by pipetting 60 μL of 1-Step Ultra TMB (Thermofisher, Waltham, MA) to the surface of the electrode and placing the plate into the Amperial™ reader where current is measured at -200 mV for 60 seconds. The current in nA is measured 3 times for each well. The process for reading the entire 96 well plate requires approximately 3 minutes.

## Plasma quantitative Amperial™ assay for SARS-CoV-2 IgG

The protocol is similar to the Amperial™ SARS-CoV-2 IgG antibody for saliva samples. Following the application of the polymer layer, 30 μL of plasma diluted at a 1:100 ratio in Casein/PBS (Thermo-Fisher, Waltham, MA) was pipetted into each well and incubated for 10 minutes at room temperature. The standard curve for plasma contains the following points: 300 ng / ml, 150 ng / ml, 75 ng / ml, 37.5 ng / ml, 18.75 ng / ml, and 0 ng / ml.

## Plasma SARS-CoV-2 ELISA assay

We purchased FDA EUA ELISA kits EUROIMMUN Anti-SARS-CoV-2 ELISA Assay for detection of IgG antibodies (EUROIMMUN US, Mountain Lakes, NJ, Product ID EI 2606–9601 G, Lot #E2001513BK). We processed samples exactly as described in the package insert.

## Human subjects

Volunteers, with prior positive qPCR tests for COVID-19 infection or positive antibody tests using currently available FDA EUA-cleared antibody tests were consented via a written consent. Subjects enrolled were all over the age of 18. Subject participants responded to a questionnaire regarding severity of symptoms, onset of symptoms, and method of diagnosis (UCLA IRB #06-05-042). Severity of symptoms were self-graded on the following 7-point scale:

    0: Asymptomatic
    1: Mild (Barely noticed, perhaps slight fever and cough)
    2: Moderate (felt moderately ill but did not need to seek medical care)
    3: Sought medical Care but was not admitted to hospital

4: Hospitalized

5: Admitted to ICU

6: Placed on Ventilator

A set of 13 paired saliva and plasma samples were provided by the Orasure™ Company.

## Saliva collection

All COVID-19 samples were obtained using the Orasure™ FDA-cleared saliva collection device and used according to manufacturer instructions. The Orasure™ collection device consists of an absorbent pad on the end of a scored plastic wand. The individual places the pad between cheek and gum for a period of 2–5 minutes. Subsequently the wand and pad are placed into a tube containing transport medium, the top of the stick is broken off, and the tube is sealed for transport. The sealed tube is placed into a zip-lock bag and shipped by any standard method. According to the package insert, samples are stable at ambient temperature for 21 days (see results below and Orasure™ website). An alternate sample collection method involves the individual swabbing the pad 4 times in the gingival tooth junction prior to placing the pad between the cheek and gum. This method has been shown to improve IgG yield in some patients with low antibody levels (personal communication with Orasure Technologies, Inc.).

## Participant recruitment method

Positive samples (determined either through a positive SARS-CoV-2 viral test or antibody test) were acquired beginning May 2020 to July 2020 via the described Orasure™ Oral Fluid Collection Device Kit previous described. Subjects were recruited into the study via electronic correspondence during the early stages of the COVID-19 pandemic in regions affected by COVID-19 (California, Illinois, New York, New Jersey). Subjects are all over the age of 18. Subjects are not representative of the general population.

Samples collected pre-2012 were used as controls. Saliva was collected from healthy individual volunteers at meetings of the American Dental Association between 2006 and 2011. Consent was obtained under IRB approval UCLA IRB #06-05-042. Both male and females, mostly non-smokers, 18–80 years of age, and differing ethnicities were included. All subjects were consented prior to collection. Each subject expectorated ~ 5 mL of whole saliva in a 50cc conical tube set on ice. The saliva was processed within 1/2 hour of collection. Samples were spun in a refrigerated centrifuge at 2600 X g for 15 minutes at 4˚C. The supernatant (cell-free saliva) was then pipetted into two-2 mL cryotubes and 1.1 μL Superase-In (Ambion, Austin, TX) was added as a preservative. Each tube was inverted to mix. The samples were frozen in dry ice and later stored in -80˚C.

## Sample size and statistical methods

Due to the nature of the pandemic and the evolving nature of EUA diagnostics during the early phases of the pandemic, no power calculations were performed for study size but instead the FDA/EUA recommendation of 30 subjects was followed. For components of work that required comparisons between groups, student's T-test was conducted. $p$ value, 0.05, corresponds to a 95% confidence or $p$ value, 0.01, corresponds to 99% confidence. Data analysis performed was using GraphPad Prism 8.2.1.

## Results

### Linearity

Fig 2 demonstrates the dynamic range and linearity of the assay. In these experiments varying amounts of monoclonal human anti-S1 IgG was added to a saliva sample from a healthy

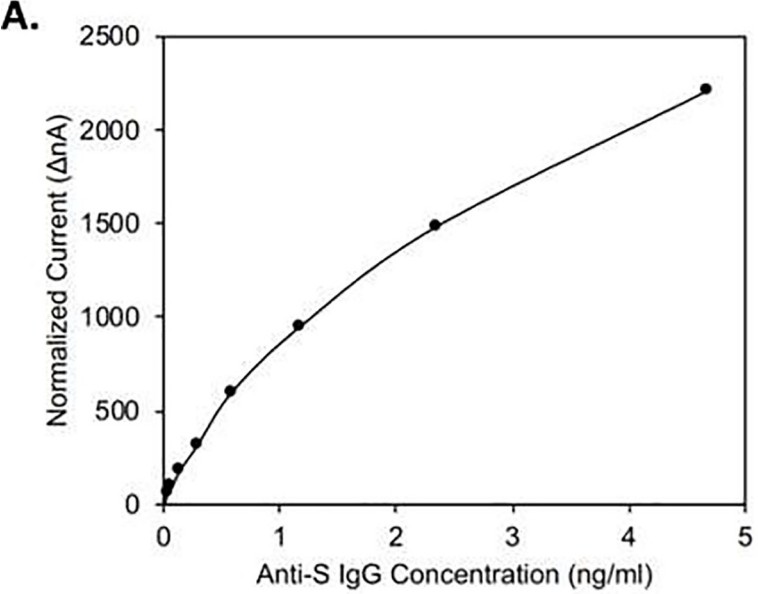

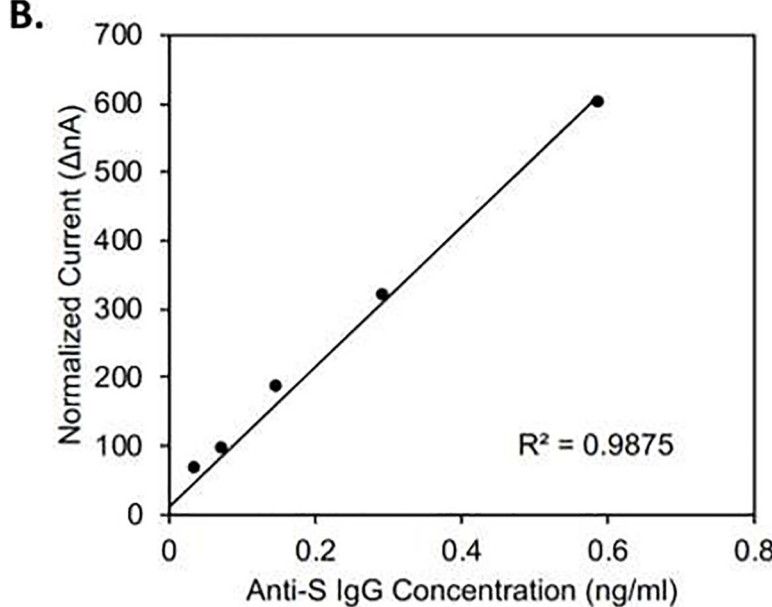

**Fig 2. Dynamic range and linear range of Amperial™ anti-Spike S1 IgG assay.** X-axis: Amount of spike in anti-SARS-CoV-2 IgG in ng / ml. Y-axis: Normalized current in nA. Panel A: 0–5 ng / ml Panel B: 0.1–0.7 ng / ml.

volunteer and subjected to the assay. Fig 2 shows a range of 0.2 to 6 ng/ml. The Y-axis shows nano-amperage measured (nA). The X-axis represents spike-in concentrations of IgG. The assay begins to become saturated at about 3 ng / ml. Fig 3 shows dilutions down to 0.03 ng / ml to 0.6 ng / ml and shows linearity in that range. This allows us to create a standard curve containing the following points: 3 ng / ml, 1.5 ng / ml, 0.75 ng / ml, 0.375 ng / ml, 0.1875 ng / ml, and 0 ng / ml.

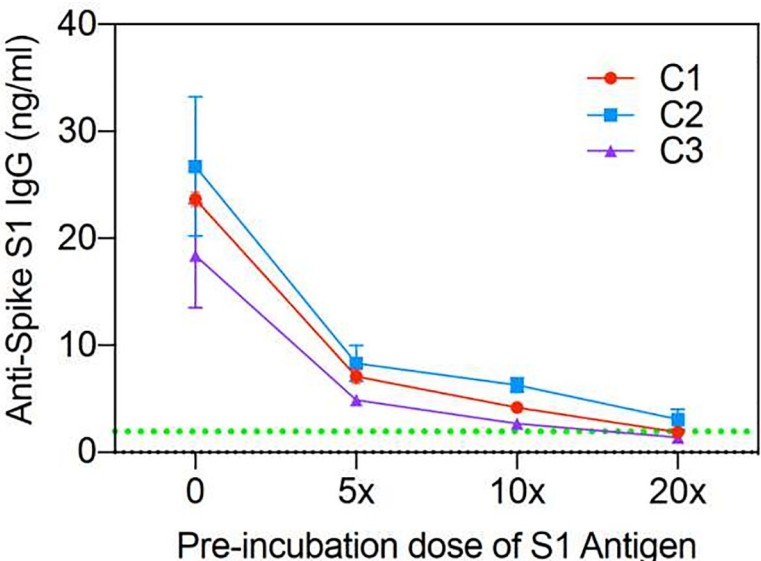

**Fig 3. Competition assay of three COVID-19 patients: C1, C2, and C3.** Varying amounts of exogenous anti-SARS-CoV-2 IgG added to saliva of 3 different recovered COVID-19 patients.

## Inhibition assay

In order to demonstrate the specificity for the assay on actual clinical samples, we used the saliva from 3 recovered patients who had high levels of SARS-CoV-2 antibodies and added exogenous S1 antigen in varying amounts prior to analysis on the Amperial™ assay. The exogenous S1 antigen should compete for binding sites and therefore extinguish the nA signal. Fig 3 shows the results of this experiment. The red, purple, and green represent 3 different patients. The X-axis demonstrates increasing concentration of exogenous S1 added to the saliva before subjecting it to the assay. As shown, saliva pre-incubated with S1 antigen extinguishes the detectable IgG signal proportionately, therefore demonstrating the specificity of the assay to S1 antigen in clinical samples.

## Matrix effects

Since we are be comparing samples collected by various methods, it is vital to determine if any significant matrix effects could interfere with data interpretation. We examined the 3 different collection methods used in this study: Expectoration/centrifugation, Orasure™ without swabbing and Orasure™ with swabbing.

Two methods of collection using the Orasure™ Oral Fluid Collection Device were tested. The first method (non-swabbing) collects saliva by placing an absorbent pad into the lower gum area for 2–5 minutes and then placing the saturated collection pad into a preservative collection tube. The second method (swabbing) adds the step of first gently rubbing the collection pad along gum line, between the gum and cheek, 5 times, before placing the device in the lower gum area for 2–5 minutes, and then immersing the saturated collection pad into the collection tube. Healthy donors (n = 5) collected their saliva using these two different methods. The control pre-2012 samples were collected with an expectoration protocol for whole saliva collection (falcon tubes), processing (centrifuge), stabilization, and storage. Five samples collected by each of the 3 methods and were analyzed in duplicate. The results are shown in Fig 4 under the heading "No spike in." There are no differences among 3 sample types. We then added monoclonal human anti-S1 IgG to each sample and again ran them in duplicate (Fig 4)

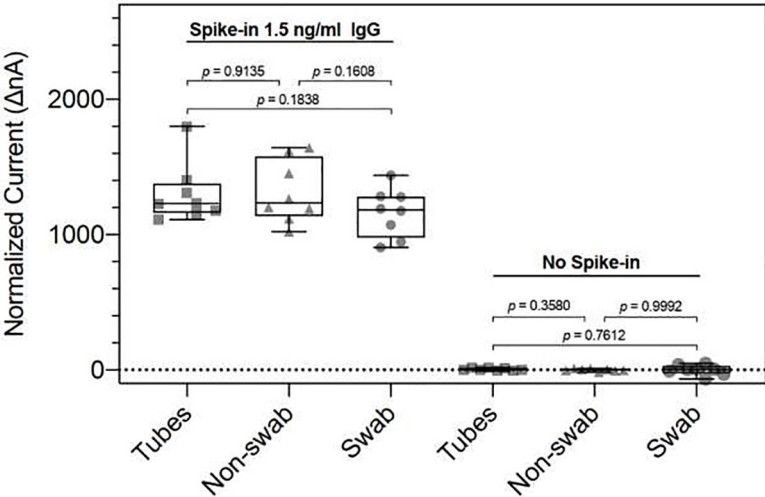

**Fig 4. Box plot of saliva matrix experiments with saliva from healthy subjects.** Green dashed line represents 5 standard deviations above the mean.

above caption Spike-in 1.5 ng / ml IgG. A non-parametric Student t-test was performed with no significant differences between any of the collection methods.

## Stability

The Orasure™ collector is an FDA-cleared device for the analysis of anti-HIV IgG. The package insert describes a 21-day stability at ambient temperature. We wished to establish the stability of anti-COVID-19 IgG using this collector. Passive whole saliva was collected from four healthy individuals using 50 mL falcon tubes and spiked with anti-Spike S1 IgG to reach a final concentration of 300 ng / ml. Aliquots of 1.75 mL of saliva were placed into 50 mL tubes and then the sponge of the Orasure™ collector was submerged into the saliva for five minutes and processed as described in Methods. The collected saliva was then aliquoted into PCR tubes and left at ambient temperature (21˚C) for 0, 1, 3, 7, and 14 days before storage at -80˚C. After 14 days, samples were thawed and assayed using the anti-Spike S1 IgG Amperial™ assay to assess stability. At 14 days, 95% of the original signal remained, demonstrating the 14-day stability of anti-SARS-CoV-2 antibodies collected in Orasure™ containers (see Fig 5).

## Specificity and reference range

Once we established no significant differences between the tube collection method and the Orasure™ collector method, we analyzed a series of 667 samples collected between 2006 and 2009 at the annual meeting of the American Dental Association. Scatter plots of these data for both nA and ng / ml are shown in Fig 6A and 6B. We established the mean and standard deviation for both raw nA values and concentration in ng / ml. In order to maximize specificity, we selected a reference range > 5 SD above the mean. A 5 sigma level would lead to a specificity of 99.9994%. In fact, we have never seen a healthy sample above the 5 sigma level. As will be seen, the sensitivity of the assay remains greater than 88% even with this rigorous specificity.

## Recovered COVID-19 patients

Fig 7 displays the scatter plot for 667 healthy controls and 34 volunteer patients who recovered from COVID-19 infection. All patients were a minimum of 14 days post onset of symptoms

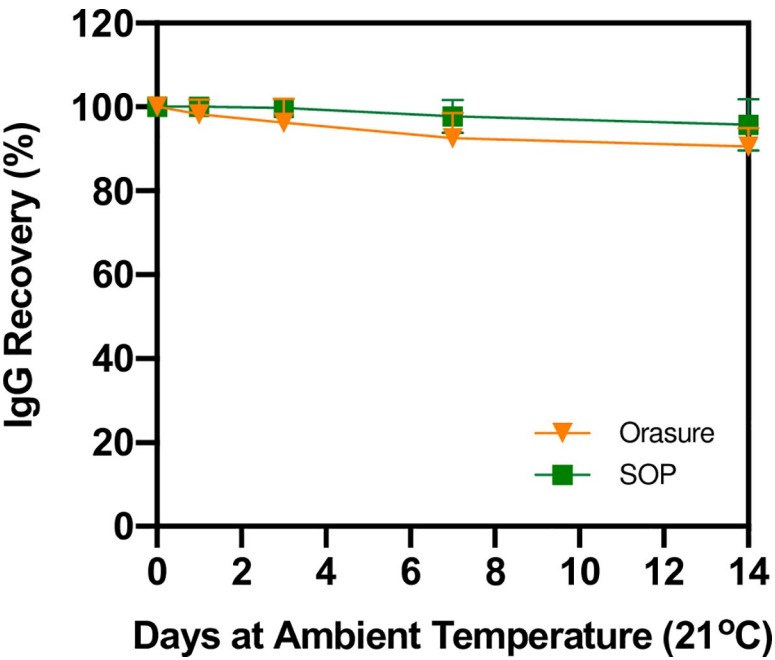

**Fig 5. Stability study performed on spike-in of SARS-CoV-2 IgG into healthy saliva specimen using two different methods (a research SOP which involves expectoration into a falcon tube and the Orasure™ Oral Fluid collection device).** The collect saliva was aliquoted and left at ambient temp for 0, 1, 3, 7, 14 days. Results were normalized relative to the measured assay signal of a sample at day 0. Results show that the sample is stable with no significant degradation for up to 14 days.

and some patients were as many as 15 weeks post symptoms. The 5 sigma cutoff is shown by the green dotted line. A more detailed discussion of the recovered patients appears in the following section. The data show that all healthy patients are negative and 30 of the 34 recovered patients are positive. These data demonstrate a sensitivity of 88% and a specificity of > 99.985%. It is important to note that not all recovered patients have detectable antibody [10] so the 4 patients with undetectable antibody may be biologically negative and not the result of lack of sensitivity of the assay.

Fig 8 demonstrates the relationship of anti-S1 IgG levels to severity of symptoms. Table 1 is a tabular summary of these data. All patients who had severity indexes ≥3 (sought medical attention, admitted to hospital, admitted to ICU, on ventilator) had positive antibody levels. Although 4 patients with mild symptoms had antibody levels in the normal range, both asymptomatic patients had appreciable antibody levels. These patients were close contacts of more severely affected patients. The highest antibody level recorded is severity index level 2 patient (moderate symptoms, did not seek medical care). It is important to note that both asymptomatic patients had easily detectable antibody levels in saliva, suggesting this test may be useful in general population screening.

## Paired saliva and plasma samples

We obtained 14 paired, blinded plasma and saliva samples. The plasma was analyzed by an FDA EUA-cleared ELISA test purchased from EUROIMMUN (see Methods). The saliva samples, collected in Orasure™ buffer, were analyzed by the Amperial™ assay described in Methods. After unblinding, we discovered 8 recovered COVID patients and 5 healthy patients in this series. All 5 healthy patients were negative in both the saliva and plasma assays. In 7 of the 8 recovered patients, both plasma and saliva tests were positive. There was one sample with a

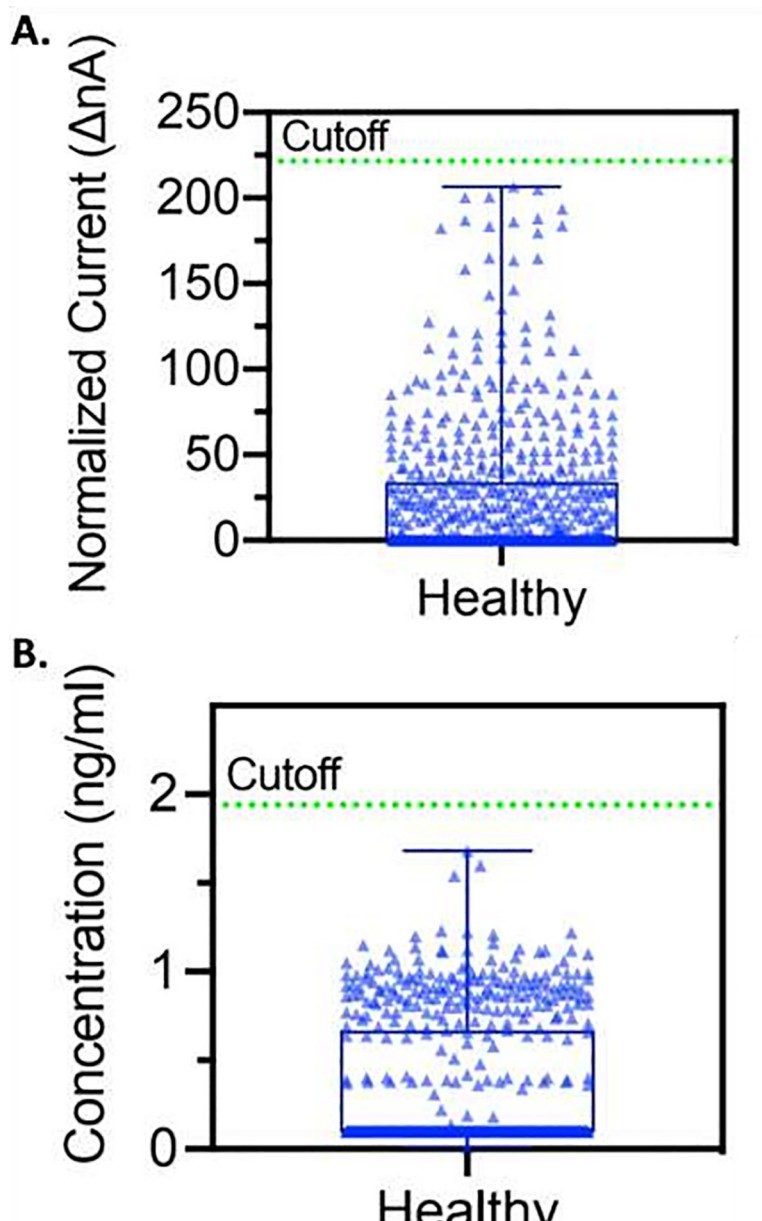

**Fig 6.** Healthy reference range of Amperial™ saliva anti-SARS-CoV-2 IgG assay of 667 unexposed subjects in (A) normalized current (ΔnA) with mean = 24.38 and cutoff = 221.47 and (B) concentration (ng / ml) with mean = 0.33 and cutoff = 1.19.

discrepancy between saliva and plasma, with the plasma positive and the saliva in the indeterminate range.

The EUROIMMUNE ELISA assay is a semi-quantitative assay and yields an absorbance ratio rather than a quantity. Fig 9 demonstrates the relationship between the saliva quantitative results and plasma absorbance ratio for the paired plasma and saliva samples. There is a clear relationship between the 2 levels, with the higher plasma absorbance ratios associated with higher saliva quantitation.

We developed a research quality assay to quantify anti-SARS-CoV-2 IgG levels in plasma (see Methods). We analyzed the 13 plasma samples using this assay. The results of this

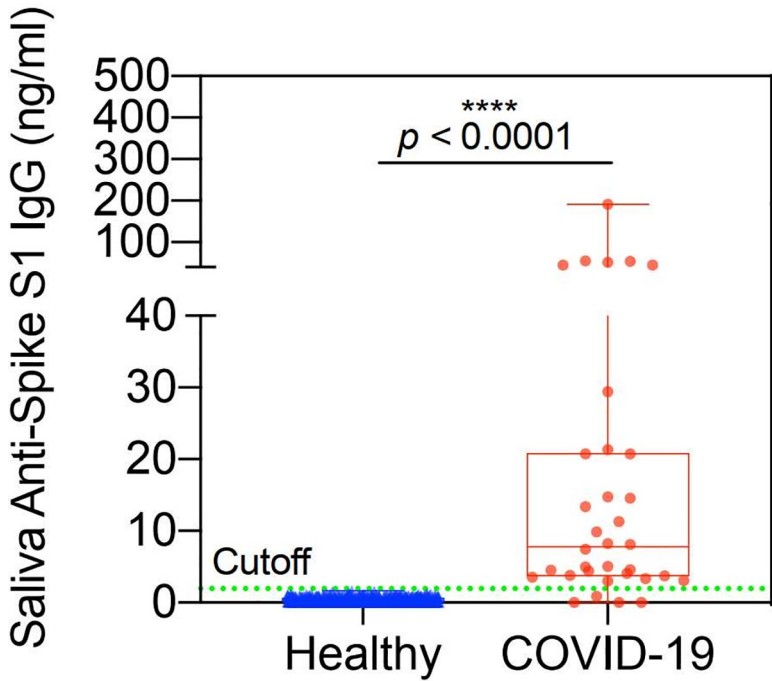

**Fig 7. Amperial™ detection of anti-Spike S1 IgG in saliva of COVID-19 (n = 34) and healthy subjects (n = 667).**
Green dashed line indicates 5 SD reference range cutoff.

experiment are shown in Fig 10. Panel A shows a log / log plot of plasma versus saliva levels showing a clustering with high plasma levels associated with high saliva levels. Panel B shows the box plot of these values, demonstrating that plasma levels are approximately 50X those of saliva. This observation explains the necessity for an extremely sensitive assay such as the Amperial™ assay in order to detect antibodies in saliva. Of note, the publication regarding saliva SARS-CoV-2 IgG detection reports levels of 25–60 mcg / ml, 1000 times less sensitive than our assay.

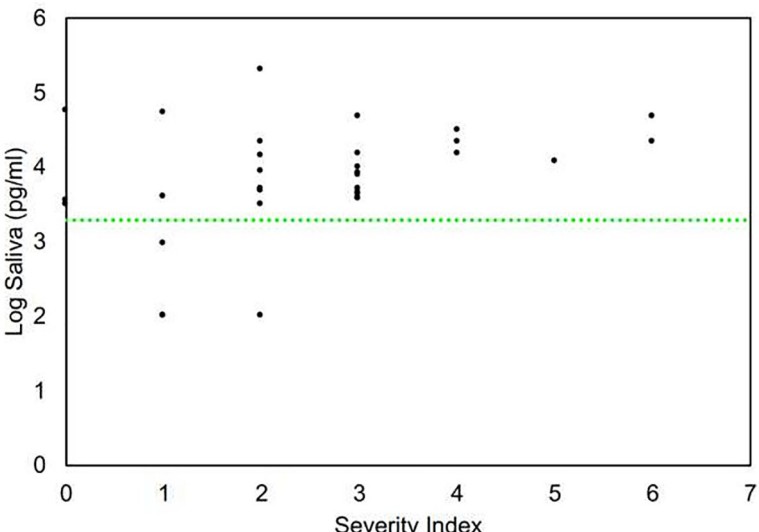

**Fig 8. Clinical severity index and anti-Spike S1 IgG level in saliva.**

**Table 1. Correlation of Amperial™ anti-SARS-CoV-2 IgG levels in saliva with severity of symptoms in 34 COVID positive subjects.**

| Severity Index | Positive IgG (no. patients) | Negative IgG (no. patients) | Sensitivity (%) |
|---|---|---|---|
| 0: Asymptomatic | 2 | 0 | 100 |
| 1: Mild Flu-Like Symptoms | 3 | 3 | 50 |
| 2: Moderate Flu-Like Symptoms | 9 | 1 | 90 |
| 3: Sought Medical Attention | 10 | 0 | 100 |
| 4: Admitted to Hospital | 3 | 0 | 100 |
| 5: Admitted to ICU | 1 | 0 | 100 |
| 6: Placed on Ventilator | 2 | 0 | 100 |
| Total | 30 | 4 | 88.24 |

## Longitudinal tracking of antibody levels

Three of our volunteers supplied samples at weekly intervals so we could determine the stability of their antibody levels. Results appear in Fig 11. The 5 standard deviation cutoff is again shown with the dashed green line. All 3 patients continued to have detectable levels for more than 12 weeks, with the longest interval of 15 weeks. All tests were positive in all patients and antibody levels in all 3 patients remained clearly positive during the time interval studied. Patients C1 and C3 seem to have a rise in antibody level between 11 and 12 weeks post initial symptoms followed by a return to baseline level. Patient C2 might also have had a spike in antibody levels at 10 weeks. This may be result of the amnestic B-cell population becoming established. There is insufficient data at this time to determine if this is a generalized pattern.

## CLIA evaluation

We performed a full CLIA laboratory developed test evaluation for the Amperial™ COVID-19 IgG Antibody test. The validation assayed 72 unaffected patients and 30 recovered patients

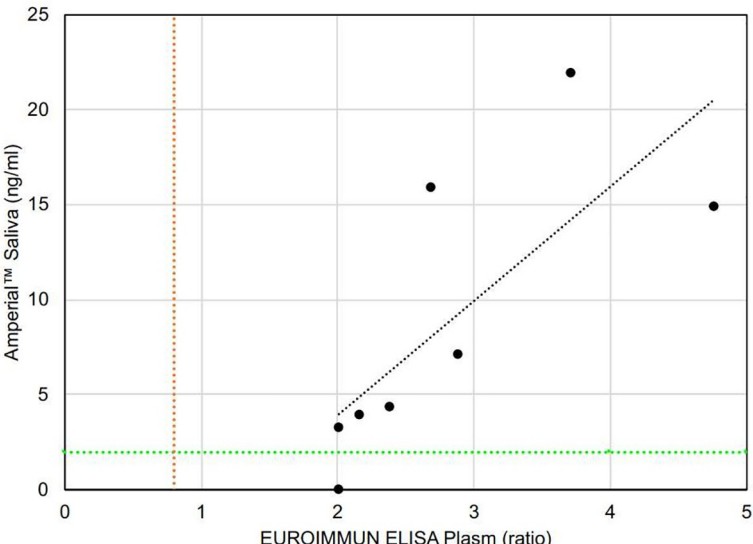

**Fig 9. COVID-19 antibody level in paired saliva and plasma of COVID-19 (n = 8) subjects in a blinded randomized cohort.** Plasma antibodies level are measured by EUROIMMUN ELISA reported in ratio (proportion of OD of calibrator to OD of sample) and saliva antibodies are measured by Amperial™ in pg / ml. Green dashed line indicates 5 SD reference range cutoff of Amperial™ test and red dashed line is reference range for EUROIMMUN ELISA.

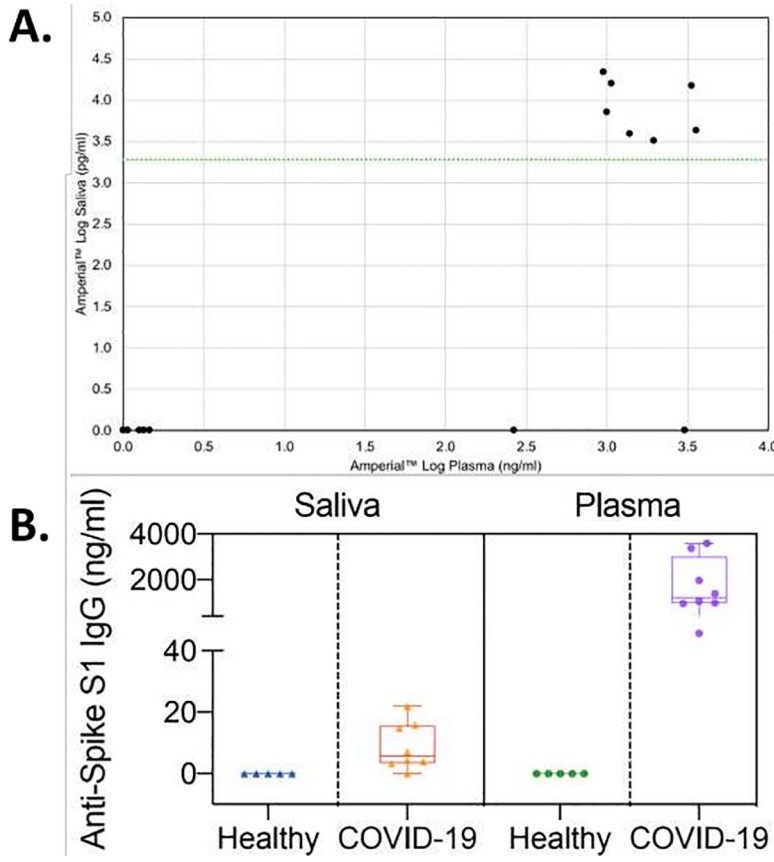

**Fig 10. Relationship of plasma anti-SARS-CoV-2 IgG levels to saliva levels measured by Amperial™ assays.** (A) Panel A shows a log / log plot of plasma versus saliva levels showing a clustering of the positive values with high plasma levels associated with high saliva levels on the Amperial™ platform. (B) Box plot of COVID-19 (n = 8) and healthy (n = 5) subjects demonstrating that the normalized plasma levels are approximately 50X those of saliva.

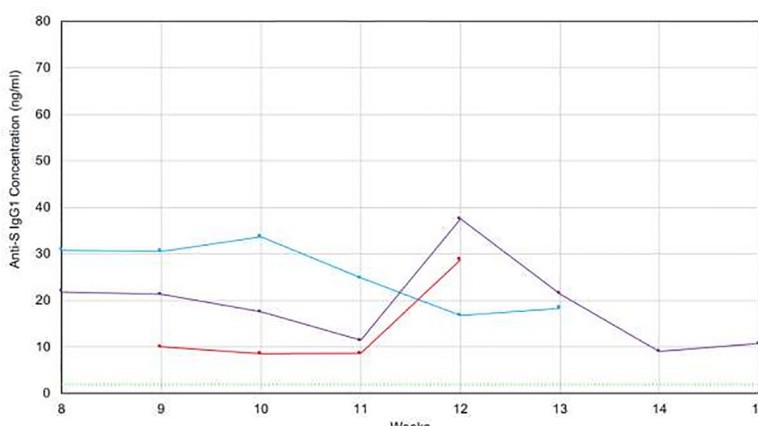

**Fig 11. Longitudinal measurement of saliva anti-SARS-CoV-2 IgG levels in 3 recovered patients.** X-axis: Time after initial onset of symptoms (in weeks). Y-axis: IgG levels measured in saliva.

and demonstrated 100% sensitivity and specificity. The intra-assay and inter-assay variability were 9.28% and 16.2% respectively.

## Discussion

We have developed an exquisitely specific, sensitive, non-invasive saliva based quantitative assay for anti-SARS-CoV-2 IgG antibodies. Our goal was to create a quantitative assay with sufficient positive predictive value to be useful to inform individuals regarding previous infection with COVID-19. By establishing a reference range of 5 sigma above than the mean we have a theoretical analytical specificity of 99.9999994%. We plan to repeat the analysis of all positive samples to further increase analytical specificity. Since our test is non-invasive with home-collection we can also offer repeat testing on a second sample to further increase specificity. These procedures will minimize the false positives due to purely technical issues. There is still the possibility of biological false positives, however, due to cross reactivity with other infectious or environmental agents. The S1 antigen appears to be specific for SARS-CoV-2 [2, 3, 10] and in our series of 667 samples collected prior to 2019 we observed no false positive results.

We cannot predict the eventual clinical specificity of this assay. At a minimum, the specificity is 667 / 668 or 99.985% assuming the next control sample tested would be a false positive, but the specificity is likely to be higher. Our current sensitivity is 100% for patients with symptoms severe enough to seek medical care. For all patients, including mildly asymptomatic patients, our clinical sensitivity is 88%. Since the Amperial™ assay only requires 6 μL of collection fluid, several assays can be performed from the same sample. This allows all positives to be repeated to confirm the positive results and further increase the specificity of the assay. We will offer testing of a second, independent sample for all patients testing positive. Since saliva collection is easily be performed at home, obtaining a second sample is not difficult.

For any laboratory test, the PPV is proportional to the prevalence of positivity in the population. A recent study demonstrated a prevalence of between 4.4% to 6% in Britain [17]. Using the minimum specificity of 99.85% and a prevalence of 6% the Amperial™ saliva assay would have a minimum PPV of 96%. In contrast, a published saliva antibody detection assay reported a specificity of 98% with a similar sensitivity (89%). This specificity leads to PPV of only 69% making it an ineffective tool for population screening.

Our data demonstrate that the Imperial™ assay is appropriate for longitudinal screening of antibody levels, a particular utility in vaccine trials and in population monitoring following mass immunization. Since this assay is quantitative and levels appear to be stable with time, patients may be monitored from home at frequent intervals. If antibodies raised in response to vaccination do not include IgG antibodies to S1 antigen, it is easy to rapidly develop Amperial™ antibody tests to any antigen. This requires adding the new antigen to the pyrrole solution and does not require significant alteration of assay conditions.

A particular advantage of this assay is convenience. The Orasure™ collector is simple and easy to use and does not require professional monitoring for adequate collection. Home collection relieves the burden to an already stressed health care system. Vulnerable populations such as children and the elderly can be guided through the collection process by parents or other adults. It is possible to obtain repeat samples to confirm positives and to perform longitudinal testing since the only requirement for testing is shipping the collecting kit.

The Amperial™ IgG test is plate-based and high-throughput. An entire plate is easily processed in 2 hours, leading to rapid turnaround time once the sample enters the laboratory. There is no pre-processing of the sample required; samples are taken directly from the collection vial and placed into the assay. With standard liquid handlers, the assay may be easily

automated allowing for extremely high-throughput since the Amperial™ reader is only required for the polymerization step of less than a minute at the beginning of the assay and 3 minutes for the measurement phase at the end of the assay.

Published data [13] and our own demonstrate a correlation between blood results and saliva results indicating that the IgG present in saliva is most likely derived from the plasma through filtration. Our data shows that saliva IgG levels are approximately 50-fold less than those in plasma necessitating a highly sensitive assay in order to detect the IgG levels in saliva.

There is some discussion in the literature of the role antibody testing may have in managing the COVID-19 epidemic. Alter and Seder published an editorial in the New England Journal of Medicine arguing, "Contrary to recent reports suggesting that SARS-CoV-2 RNA testing alone, in the absence of antibodies, will be sufficient to track and contain the pandemic, the cost, complexity, and transient nature of RNA testing for pathogen detection render it an incomplete metric of viral spread at the population level. Instead, the accurate assessment of antibodies during a pandemic can provide important population-based data on pathogen exposure, facilitate an understanding of the role of antibodies in protective immunity, and guide vaccine development [14]".

## Conclusion

In this article, we describe the development of a non-invasive, home collection based, exquisitely specific, and acceptably sensitive test for the presence of anti-SARS-CoV-2 antibodies in saliva. This may be an important tool in controlling the pandemic and facilitating and understanding of the role of antibody production in COVID-19 immunity. Longitudinal monitoring of anti-SARS-CoV-2 IgG levels could also play a valuable role in vaccine development and deployment by allowing longitudinal quantitative assessment of antibody levels. If the presence of detectable anti-COVID-19 IgG is shown to be an indicator of immunity to reinfection, measurement of these antibodies could allow individuals to safely return to work, school and community. The Amperial™ SARS-CoV-2 assay fulfills the requirements for all of these applications.

## Author Contributions

**Conceptualization:** Samantha H. Chiang, Michael Tu, Jordan Cheng, Fang Wei, Feng Li, Richard Bender, Charles M. Strom, David T. W. Wong.

**Data curation:** Michael Tu, David Chia, Charles M. Strom.

**Formal analysis:** David Chia.

**Investigation:** Samantha H. Chiang, Michael Tu, Jordan Cheng, Fang Wei, Feng Li, Charles M. Strom.

**Methodology:** Samantha H. Chiang, Michael Tu, Fang Wei, Charles M. Strom.

**Project administration:** Charles M. Strom.

**Supervision:** Charles M. Strom, David T. W. Wong.

**Validation:** Omai Garner, Sukantha Chandrasekaran.

**Writing – original draft:** Charles M. Strom.

**Writing – review & editing:** Samantha H. Chiang, Michael Tu, Richard Bender, Charles M. Strom, David T. W. Wong.

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
