## [Decision Letter · Decision Letter 0]

26 Mar 2021

PONE-D-21-01466

Development and validation  of a highly sensitive and specific electrochemical assay to quantify anti-SARS-CoV-2 IgG antibodies to facilitate pandemic surveillance and  monitoring of vaccine response.

PLOS ONE

Dear Dr. Storm

Thank you for submitting your manuscript to PLOS ONE. After careful consideration, we feel that it has merit but does not fully meet PLOS ONE’s publication criteria as it currently stands. Therefore, we invite you to submit a revised version of the manuscript that addresses the points raised during the review process. There are minor comments from both expert reviewers, please address them ASAP and resubmit again as your assays is very time sensitive. 

We look forward to receiving your revised manuscript.

Kind regards,

Gheyath K. Nasrallah, PhD

Academic Editor

PLOS ONE

Journal Requirements:

2. In your Methods section, please provide additional information about the participant recruitment method and the demographic details of your participants. Please ensure you have provided sufficient details to replicate the analyses such as:

a) the recruitment date range (month and year),

b) a description of any inclusion/exclusion criteria that were applied to participant recruitment,

c) a table of relevant demographic details,

d) a statement as to whether your sample can be considered representative of a larger population,

e) a description of how participants were recruited, and f) descriptions of where participants were recruited and where the research took place.

3. Please provide a sample size and power calculation in the Methods, or discuss the reasons for not performing one before study initiation.

4. To comply with PLOS ONE submission guidelines, in your Methods section, please provide additional information regarding your statistical analyses. For more information on PLOS ONE's expectations for statistical reporting, please see https://journals.plos.org/plosone/s/submission-guidelines.#loc-statistical-reporting

5. Please provide the product number and any lot numbers of the ELISA kit purchased from for your study.

6. Please note that PLOS does not permit references to “data not shown.” Authors should provide the relevant data within the manuscript, the Supporting Information files, or in a public repository. If the data are not a core part of the research study being presented, we ask that authors remove any references to these data.

7. Please provide additional details regarding participant consent.

In the ethics statement in the Methods and online submission information, please ensure that you have specified what type you obtained (for instance, written or verbal, and if verbal, how it was documented and witnessed).

If your study included minors, state whether you obtained consent from parents or guardians.

If the need for consent was waived by the ethics committee, please include this information.

8. Thank you for stating the following in the Competing Interests section:

'I have read the journal's policy and the authors of this manuscript have the following competing interests: MT is an employee of Liquid Diagnostics, LLC (LD) and is an equity holder. CS and RB are founders and board members of LD and are equity holders. DW is an equity holder in LD.'

We note that one or more of the authors are employed by a commercial company: Liquid Diagnostics, LLC (LD)

8a. Please provide an amended Funding Statement declaring this commercial affiliation, as well as a statement regarding the Role of Funders in your study. If the funding organization did not play a role in the study design, data collection and analysis, decision to publish, or preparation of the manuscript and only provided financial support in the form of authors' salaries and/or research materials, please review your statements relating to the author contributions, and ensure you have specifically and accurately indicated the role(s) that these authors had in your study. You can update author roles in the Author Contributions section of the online submission form.

8b. Please also provide an updated Competing Interests Statement declaring this commercial affiliation along with any other relevant declarations relating to employment, consultancy, patents, products in development, or marketed products, etc. 

8c. Please include both an updated Funding Statement and Competing Interests Statement in your cover letter. We will change the online submission form on your behalf.

9. Please amend either the title on the online submission form (via Edit Submission) or the title in the manuscript so that they are identical.

Reviewers' comments:

Reviewer's Responses to Questions

**Comments to the Author**

1. Is the manuscript technically sound, and do the data support the conclusions?

Reviewer #1: Yes

Reviewer #2: Yes

2. Has the statistical analysis been performed appropriately and rigorously? 

Reviewer #1: I Don't Know

Reviewer #2: Yes

3. Have the authors made all data underlying the findings in their manuscript fully available?

Reviewer #1: Yes

Reviewer #2: Yes

4. Is the manuscript presented in an intelligible fashion and written in standard English?

Reviewer #1: Yes

Reviewer #2: Yes

5. Review Comments to the Author

Reviewer #1: The initial sentence of the abstract, and the rest of the abstract, do not clearly disclose how the assay works. What compound is electrochemically detected?

It needs careful proofreading. E.g.,SARS-CoV-2 is misspelled in the abstract and Fig 9, and Line 57 should say "development *of* symptoms".

 Are the references in line 99 correct?

 24 washes are used in the assay, a rather large number. Were so many required?

 With whom was the communication cited in line 186?

Some of the controls were 14 years old. Is there any evidence for the stability of antibodies in these samples?

In lines 328 and 329, the receptor binding domain is described by 2 different incorrect names.

The number of paired saliva and plasma samples is rather small. Have these numbers increased since the preprint or manuscript were prepared?

The impressively-measured saliva antibody levels seem high enough for standard ELISA.  Why use a specialized instrument?

Reviewer #2: It is interesting work that could be useful to detect SARS-CoV-2 antibodies in saliva samples for COVID-19 diagnostics and for longitudinal monitoring of antibody levels in recovered patients. It is not entirely clear how easy/challenging sample processing/preparation is, what the level of reader complexity is, how long sample processing takes, etc. More details on underlying technology would be useful. Statements about theoretical "99.9999994%" specificity are somewhat dubious and not very meaningful.

6. PLOS authors have the option to publish the peer review history of their article (what does this mean?). If published, this will include your full peer review and any attached files.

Reviewer #1: No

Reviewer #2: No

---

## [Author Response · Author response to Decision Letter 0]

1 Apr 2021

We wish to thank the reviewers and editor for their time. We have attached a file titled "Response to Reviewers" containing a response to the various items requested by the editor and reviewers.

---

## [Decision Letter · Decision Letter 1]

26 Apr 2021

Development and validation of a quantitative,non-invasive, highly sensitive and specific, electrochemical assay for anti-SARS-CoV-2 IgG antibodies in saliva

PONE-D-21-01466R1

Dear Dr. Strom,

We’re pleased to inform you that your manuscript has been judged scientifically suitable for publication and will be formally accepted for publication once it meets all outstanding technical requirements.

Kind regards,

Chandrabose Selvaraj, Ph.D.

Academic Editor

PLOS ONE

Additional Editor Comments (optional):

Reviewers' comments:

Reviewer's Responses to Questions

**Comments to the Author**

1. If the authors have adequately addressed your comments raised in a previous round of review and you feel that this manuscript is now acceptable for publication, you may indicate that here to bypass the “Comments to the Author” section, enter your conflict of interest statement in the “Confidential to Editor” section, and submit your "Accept" recommendation.

Reviewer #2: All comments have been addressed

Reviewer #3: (No Response)

2. Is the manuscript technically sound, and do the data support the conclusions?

Reviewer #2: Yes

Reviewer #3: (No Response)

3. Has the statistical analysis been performed appropriately and rigorously? 

Reviewer #2: Yes

Reviewer #3: (No Response)

4. Have the authors made all data underlying the findings in their manuscript fully available?

Reviewer #2: Yes

Reviewer #3: (No Response)

5. Is the manuscript presented in an intelligible fashion and written in standard English?

Reviewer #2: Yes

Reviewer #3: (No Response)

6. Review Comments to the Author

Reviewer #2: (No Response)

Reviewer #3: In this paper, the authors report an electrochemical anti-Spike S1 IgG assay. The assay show good sensitivity and selectivity for IgG antibody detection in saliva samples from patients recovered from COVID-19. Overall it is a nice and convincing work and I would like to recommend publication in PLOS One.

7. PLOS authors have the option to publish the peer review history of their article (what does this mean?). If published, this will include your full peer review and any attached files.

Reviewer #2: No

Reviewer #3: No

---

## [Editor Report · Acceptance letter]

23 Jun 2021

PONE-D-21-01466R1 

Development and validation  of a quantitative,non-invasive, highly sensitive and specific, electrochemical assay for anti-SARS-CoV-2 IgG antibodies in saliva 

Dear Dr. Strom:

I'm pleased to inform you that your manuscript has been deemed suitable for publication in PLOS ONE. Congratulations! Your manuscript is now with our production department. 

Kind regards, 

on behalf of

Dr. Chandrabose Selvaraj 

Academic Editor

PLOS ONE